# One-Pot Synthesis and Surfactant Removal from MCM-41 Using Microwave Irradiation

**DOI:** 10.3390/molecules29020460

**Published:** 2024-01-17

**Authors:** Marília R. Oliveira, Yasmin T. Barboza, Thauane S. L. Silva, Juan A. Cecilia, Enrique Rodríguez-Castellón, Silvia M. Egues, Juliana F. De Conto

**Affiliations:** 1Center for Studies in Colloidal Systems (NUESC), Laboratory of Materials Synthesis and Chromatography, Institute of Technology and Research (ITP), Aracaju 49032-490, SE, Brazil; marilia-oliveiras@hotmail.com (M.R.O.); yasminteless@hotmail.com (Y.T.B.); thauane_selva@hotmail.com (T.S.L.S.); smsegues@gmail.com (S.M.E.); jfconto@gmail.com (J.F.D.C.); 2Postgraduate Programme in Process Engineering, Tiradentes University (UNIT), Aracaju 49032-490, SE, Brazil; 3Department of Inorganic Chemistry, Crystallography, and Mineralogy, Faculty of Sciences, University of Malaga, 29071 Málaga, Spain; castellon@uma.es

**Keywords:** MCM-41 silica, one-pot synthesis, CTAB, microwave irradiation

## Abstract

This research pioneers the application of microwave irradiation as an innovative strategy for one-pot synthesis and surfactant elimination (cetyltrimethylammonium bromide—CTAB) from MCM-41, introducing a rapid and efficient methodology. MCM-41 silica is widely utilized in various applications due to its unique textural and structural properties. Nonetheless, the presence of residual surfactants after synthesis poses a challenge to its effective application. MCM-41 synthesis, conducted in a microwave reactor at 60 °C, provided a result within 0.5 to 1 h. Comprehensive analyses of structural, chemical, morphological, and surface characteristics were undertaken, with a focus on the impact of synthesis time on these properties. Surfactant extraction involved the use of ethanol as a solvent at 120 °C for 6 min within the microwave reactor. The acquired particles, coupled with the properties of textural and structural features, affirmed the efficacy of the synthesis process, resulting in the synthesis of MCM-41 within 36 min. This study presents the first instance of one-pot synthesis and surfactant removal from MCM-41 using a microwave reactor. The proposed method not only addresses the surfactant removal challenge, but also substantially accelerates the synthesis process, thereby enhancing the potential for MCM-41’s application in diverse fields.

## 1. Introduction

In the ever-evolving landscape of materials science, this paper represents a significant advance by presenting an innovative approach for the synthesis of MCM-41 silica through microwave irradiation. By delving into this cutting-edge technology, this study seeks not only to optimize the synthesis process, but also to highlight the substantial benefits that this emerging methodology offers. Mesoporous silica MCM-41 was reported in 1992 by scientists at Mobil Oil Corporation and has been gaining in prominence in the scientific world due to its physicochemical properties, such as uniform pore size, thermal stability, large specific surface area, and easy modification of hydroxyl groups. The synthesis is carried out under hydrothermal conditions using a silica source and an alkyl trimethylammonium surfactant [1]. However, this surfactant is retained in the silica pores when silica is formed and must be removed after synthesis to generate the material’s porous structure.

Several techniques for surfactant removal are reported in the literature, such as calcination [2], Soxhlet extraction [3], microwave-assisted extraction [4], extraction with supercritical fluid [5], extraction with ultrasound [6], and ion exchange [7], among others. The calcination process holds precedence and takes place post-synthesis, entailing subjection of the silica to elevated temperatures within a furnace to remove the surfactant. Nevertheless, this methodology demands the utilization of elevated temperatures and extended durations (500 °C and 5 h). To reduce the temperature and time, mitigating the energy consumption in the template removal, the utilization of microwave irradiation (MW) emerges as a more efficient technique within the synthesis and elimination processes [8,9].

Jabariyan and Zanjanchi [10] carried out a procedure for the removal of CTAB from mesoporous silica MCM-41 using ultrasound. The extraction conditions were varied, with extraction time spanning 1–60 min, and temperature ranging 25–60 °C. Most surfactant molecules (93%) were removed in a 15 min sonication at 40 °C. Cheng et al. [11] synthesized MCM-41 silicas using microwave irradiation, varying different conditions such as time, power, and temperature. The authors reported that microwaves can promote the dissolution and recrystallization of the precursor and accelerate the crystallization rate, in addition to obtaining better results than synthesis carried out using conventional heating. However, the surfactant removal process was carried out through calcination. The experimental results indicated that the ideal synthesis conditions for MCM-41 were 15 min at 100 °C and 200 W, followed by calcination for 5 h at 500 °C.

Bordoni et al. [12] used microwave irradiation for pore expansion of uncalcined mesoporous materials based on SiO_2_ (core-shell and SBA-15 nanoparticles). The use of microwaves was shown to significantly decrease the reaction time of this process and to be compatible with both ionic and non-ionic surfactant pore models. The post-treatment with microwaves of calcined mesoporous materials with pore-expanding agents allowed the control of and increase in the final dimensions of the pores of the material, as well as the adaptation of the molecular structure of the materials. García-Uriostegui et al. [2] synthesized MCM-41-type nanoparticles with microwave irradiation (10 min) synthesis and removed the surfactant using the calcination method (4 h—550 °C). Silica nanoparticles with a high surface area (S_BET_ ~1000 m^2^ g^−1^) were obtained and the predominant spherical morphology was maintained. They noticed a great reduction in synthesis time and that microwave irradiation allowed the generation of a greater amount of silanol groups, which may be more effective in biomedical and catalytic applications.

These studies provided simpler and favorable methodologies for surfactant removal and MCM-41 synthesis. However, there is a clear advantage when using microwave irradiation to obtain MCM-41, where the total time required for synthesis is considerably shorter over a wide range of conditions. In comparison to conventional methods, in addition to using a greater amount of solvents and higher temperatures, a minimum of 10 h is required to synthesize the ordered silica from the MCM family [1,11]. Furthermore, in processes involving microwave irradiation, energy is provided through the interaction of the material’s electronic structure with the alternating electric field, providing energy where it is really needed. This will provide a more homogeneous temperature distribution in the reaction system, resulting in greater uniformity in pore formation in MCM-41.

As reported, many researchers have already reported the synthesis of MCM-41 as well as the removal of the surfactant with microwaves [7,11]; however, there are still no studies that carry out these two steps via one-pot synthesis. In this work, to further accelerate the silica formation process, in addition to the synthesis of MCM-41 with MW, the CTAB surfactant was removed with MW in a single step (one-pot). Therefore, the relevance of these accelerated processes in obtaining silicas lies in their ability to drive innovation, improve product quality, and promote advances in various industrial fields.

## 2. Results and Discussion

In Figure 1, through N_2_ adsorption–desorption isotherms at −196 °C, it is evident that the silicas demonstrated a type IV(a) isotherm, signifying the sorption characteristics that are characteristic of MCM-41 mesoporous silica. This is characterized by non-intersecting adsorption and desorption segments, which exhibit H4-type hysteresis. The H4 hysteresis within the desorption segment confirms the presence of mesopores, showing a sharp inflection at a relative pressure P/P_0_ of 0.5–0.9 [12,13]. The pore size distribution by BJH presents similar results between the two silicas, where a narrower pore distribution can be observed.

For comparison, the structural properties of MCM-41 synthesized in an autoclave for 24 h and calcined at 550 °C for 6 h were also analyzed (Appendix A). The figure shows a type IV isotherm, characteristic of this material, and shows an improvement in the surface area; however, its preparation time is very long. The pore size distribution is also consistent with the samples synthesized in the microwave reactor.

Regarding the conditions used in the microwave reactor, it is noticed that with the decrease in irradiation time, there is an increase in surface area (S_BET_) and pore volume (Vp) (Table 1) with favorable results, since it is possible to obtain silicas with good textural properties in lower time (0.5 h). Note that the Vps found are similar between the samples (0.72 to 0.89 cm^3^ g^−1^). It is known that MCM-41 silica has higher surface area values; however, even with this decay, the values are following values reported in the literature.

Jabariyan and Zanjanchi [10] synthesized MCM-41 silica using the conventional method for 2 h and carried out surfactant removal using different methods, such as magnetic stirring with ethanol/methanol, sonication with ethanol, and calcination in a muffle furnace. Surface area values varied between 700 and 1300 m^2^ g^−1^. Oliveira et al. [13] carried out synthesis of MCM-41 for 1 h at 60 °C (Table 1) and 80 °C in a CEM microwave reactor and calcined in a muffle for 6 h at 550 °C, obtaining an S_BET_ of 999 and 744 m^2^ g^−1^, respectively. Cazula et al. [14] synthesized MCM-41 varying the silica sources (TEOS and rice husks) for 2 h, calcining in a muffle at 520 °C for 6 h. The authors obtained S_BET_ between 200 and 1300 m^2^ g^−1^. Both studies aimed to produce MCM-41, and showed similar surface areas; however, they required a longer time for synthesis and removal of the surfactant.

It is worth mentioning that it would be possible to obtain even better textural and structural properties, since this decrease in surface area is related to the amount of surfactant still present on the silica surface, causing a blockage in the pores. This pore blockage can be associated with a decrease in micropore volume (V_mp_) from 0.59 to 0.15 cm^3^ g^−1^ when comparing the removal of CTAB using calcination in a muffle furnace (MCM-41-M60-1) with removal using microwave irradiation (MCM-41-OP-0.5), suggesting the presence of the surfactant in the silica micropores.

The CHN elemental analysis confirms this statement, as the values obtained (14.1% and 15.2%—Table 1) show that the percentage of carbon is still expressive, which is not normal for pure silica. A pure silica MCM-41 synthesized in a microwave reactor at 60 °C for 1 h and calcined in a muffle at 550 °C—6 h (MCM-41-M60-1) presents an average percentage of carbon of 0.22% [13]. This shows that CTAB is still present on the surface of the silica synthesized in this work.

In Figure 2, the infrared spectrum shows the results obtained for one-pot MCM-41 silicas, which corroborates the results obtained using the N_2_ adsorption isotherm and CHN elemental analysis. As a comparative analysis, MCM-41 silica synthesized at 60 °C for 1 h in a microwave reactor and calcined in a muffle (MCM-41-M60-1) and uncalcined MCM-41 (MCM-41-CTAB) are also presented.

The spectrum displays the absorption bands, referring to the fundamental vibrations of the SiO_2_ lattice around 800 and 1045 cm^−1^ that are characteristic of the Si-O-Si and Si-OH groups [1,9]. MCM-41-OP-0.5 and MCM-41-OP-1 silicas also showed a more intense band in the region of 960 cm^−1^ that is attributed to Si-OH stretching, showing that silica presents a visible band of silanol groups [15,16]. It is verified that through microwave irradiation it was not possible to completely remove the CTAB from the silica, being visible in the region of 1473 and 2900 cm^−1^. The efficiency of the surfactant extraction depends on the solvent used and the strength of the interaction between the organic molecules and the structure. Furthermore, it is worth evaluating different agitation times and temperature conditions, as they are essential factors in this process and can result in more efficient removal. It is believed that the possibility of total CTAB removal may involve several consecutive extraction steps with solvent switching [10].

Jabariyan and Zanjanchi [10] evaluated the effect of solvents (ethanol and methanol) and sonication/agitation on the removal of CTAB from MCM-41 silica. The authors reported the best removal results with ultrasound sonication using ethanol during two extraction cycles for 15 min at 40 °C. The authors were able to remove 93% of the surfactant, compared to muffle-fired silica. In this study, an attempt was made to extract CTAB in a microwave reactor; however, it is believed that because it has a higher dielectric constant than ethanol, the reactor overheated, making it impossible to complete the tests. Still, it should be noted that the dielectric constant of a substance varies with temperature, and this dependence is not yet known or been tabulated for many compounds, which can complicate matters [17].

It is noteworthy that for MCM-OP-0.5 synthesized for 30 min, the presence of CTAB is lower than in MCM-41 synthesized for 1 h (MCM-41-OP-1), showing that with a shorter synthesis time, it was already possible to obtain silica with a smaller amount of surfactant. This result is also confirmed in the thermogravimetric analysis, shown in Figure 3.

The mass loss was reported in three stages, showing the first loss (1^st^) up to 150 °C, which can be attributed to the water desorption. The second loss (2^nd^) between 150 °C and 320 °C (34.67%) was attributed to the decomposition of the CTAB, which is not seen in the MCM-41-M60-1 since this sample was calcined in a muffle (550 °C—6 h). It is possible to observe that the mass loss of the MCM-41-OP-0.5 is lower than that of the MCM-41-OP-1 silica, showing that the CTAB had a higher removal in this condition. The greatest loss in this range is attributed to silica that has not gone through an extraction step or surfactant calcination. The last loss (3^rd^) between 320 °C and 500 °C corresponded to a small amount of dehydroxylation of the surface of mesoporous materials [5,7].

XRD and TEM were employed to confirm if the ordering and structure of the silica channels were damaged with the one-pot microwave methodology. Through the X-ray diffraction profile (Figure 4), the presence of the main signal—allowed range 2θ = 0.5 to 3°—already suggests the hexagonal mesoporous structure of this silica [3,18,19]. It is worth mentioning that in the synthesis and removal of the surfactant using the one-pot methodology, the time was greatly reduced in the microwave reactor and it is possible that the shorter radiation time may have resulted in the formation of “hot spots” [15] influencing the ability to obtain a less defined ordering of the structure, where it would be possible to better visualize the signals.

Furthermore, it should be noted that the MCM-41-OP-1 silica, which had a longer synthesis time in the microwave reactor, presented a lower intensity peak, which may be associated with its textural properties, such as a decrease in surface area and silica ordering, suggesting that the shortest synthesis time (0.5 h) would already be a more propitious condition for one-pot synthesis. In addition, using the values obtained in the (100) plane, it was possible to calculate the parameter a0, referring to the distance between the centers of two adjacent pores (a_0_). It is noted that for MCM-41-OP-1 there was a small increase, which is associated with changes in the pore wall thickness and/or pore size, corroborating the results obtained from the N_2_ adsorption/desorption analysis [9]. Thus, combining the one-pot methodology with microwave irradiation also provides materials with a well-ordered structure [20], which can be confirmed through the images obtained with TEM, shown in Figure 5.

The formation of aligned and hexagonal channels, characteristic of MCM-41 silica, was confirmed with transmission electron microscopy. The configuration in the form of equidistant parallel lines was also verified [9,21]. However, in Figure 5b, in the upper right part of MCM-41-OP-1, there is a visualization of a black zone, which is related to a greater amount of CTAB in the silica pores [22]. In the MCM-41 that was calcined in an autoclave (Appendix A), there is no predominance of these black areas because the surfactant has been completely eliminated. It is known that the MW synthesis process is different from the conventional heating methodology; while in microwave irradiation, it is possible to generate an accelerated crystallization rate in a short period of time [23], in the case of one-pot synthesis, it is possible that the mesoporous structure and the ordered particles were generated in the initial minutes when the microwaves are being transmitted at higher power.

Additionally, through SEM (Figure 6) it was possible to verify that the silicas presented a similar morphology to the MCM-41 silica particles synthesized in a microwave reactor and calcined in a muffle, proving that the short microwave irradiation time did not have a significant impact on their morphology. The silicas presented a morphology like fibers and an agglomerate of rounded particles, and these results agree with the properties of MCM-41 [7,24].

The results obtained so far already appear to be quite promising and present textural properties consistent with the results found in the literature. Table 2 presents a bibliographic survey of studies synthesizing MCM-41 silicas with microwaves and performing surfactant removal using different methods. The shorter synthesis + removal time reported by Cheng et al. [11] is equivalent to 315 min (15 min of MW synthesis + 5 h of calcination in the muffle). There have been no reports yet that have carried out the synthesis and extraction of the surfactant in one step; therefore, the results obtained from the tests carried out suggest that the one-pot synthesis presents an effective methodology, as the methodology is in good agreement with the results found in the literature.

In short, the main purpose of this study was to minimize the time and energy used in synthesizing porous silica and removing the CTAB surfactant from MCM-41 silica, while maintaining the properties of MCM-41, which was accomplished. Compared to other surfactant removal methods that require a greater amount of time and solvents, such as calcination, ion exchange, extraction with Soxhlet, supercritical fluid, and ultrasound [2,6,7], as well as conventional synthesis with an autoclave involving long reaction times, it is evident that microwave irradiation presented significant advantages in the synthesis of surfactant removal, optimizing synthesis steps and reducing the time and energy used in the process.

Furthermore, it is emphasized that the synthesis and removal of CTAB from MCM-41-OP-0.5 silica with the one-pot methodology lasted only 36 min and, as it is an innovative process, it still needs to go through optimization steps. The one-pot synthesis already appears to be an extremely efficient methodology, where the reagents undergo uninterrupted chemical reactions within just one reactor, presenting wide applications in organic and inorganic chemistry [13,26]. In addition, microwave irradiation promoted a more homogeneous and controlled temperature distribution, avoiding the formation of thermal gradients and improving the quality and uniformity of the final product.

## 3. Materials and Methods

### 3.1. Materials

For the one-pot synthesis of silica MCM-41, the following materials were used: cetyltrimethylammonium bromide—CTAB 98% (Sigma-Aldrich, São Paulo, Brazil), aqueous ammonia solution 25% (Merck, São Paulo, Brazil), tetraethyl orthosilicate—TEOS 98% (Sigma-Aldrich, São Paulo, Brazil). To remove the CTAB surfactant, the solvent ethanol was used 99% (Sigma-Aldrich, São Paulo, Brazil). The microwave used in the synthesis was the CEM, single-mode type, Discovery SP model.

### 3.2. One-Pot Synthesis and Removal of CTAB Surfactant from Ordered Mesoporous Silica MCM-41 Assisted by Microwave Irradiation

The synthesis of MCM-41 silica and surfactant removal in a single step (one-pot) was carried out in the microwave reactor and is shown in Figure 7. Initially, 1.4 g of CTAB was placed in a beaker containing 60 mL of ultrapure water, under agitation, at room temperature. Then, 4.75 g of aqueous ammonia was added to the reaction, followed by the dropwise addition of 5 g of TEOS. The solution was stirred for 1 h at room temperature, then transferred to a quartz tube and inserted into the microwave reactor at 60 °C for 0.5 and 1 h. Then, the supernatant remaining in the tube was removed and the surfactant removal process was started. CTAB removal was adapted from the procedure described by Deekamwong and Wittayakun [7]. Thus, 20 mL of EtOH was added to the reactor for 3 min (2×) at a temperature of 120 °C. Then, the supernatant solvent was removed, and the samples were filtered with ethanol and dried in an oven at 100 °C for 8 h. The samples were designated as MCM-41-OP-0.5 and MCM-41-OP-1. For comparative analyses, MCM-41 was synthesized in a microwave reactor at 60 °C for 1 h (MCM-41-M60-1) and in an autoclave at 100 °C for 24 h (MCM-41-AC); then, both were calcined in a muffle at 550 °C for 6 h. A part of the uncalcined silica was also characterized and designated as MCM-41-CTAB.

### 3.3. Characterization

Textural properties: Before the measurements were taken, the samples were outgassed overnight at 120 °C and 10^−4^ mbar. The determination of the textural properties was carried out from the N_2_ adsorption–desorption isotherms at −196 °C using Micromeritics equipment, model ASAP 2420. The specific surface area was determined with the Brunauer–Emmett–Teller equation (BET) using the adsorption data in the range of relative pressures from 0 to 1 bar. The pore volume and pore size distribution were calculated using desorption branches of nitrogen isotherms with BJH. The total pore volume was calculated from adsorbed N_2_ at a relative pressure (P/P_0_) = 0.99. Elemental chemical analysis CHN: the analysis was carried out by burning the samples at 1100 °C in pure oxygen, using a CHNS EA3000 analyzer.

Fourier-transform infrared spectroscopy (FTIR): For the collection of the spectra, a regular spectral precision of 4 cm^−1^ was employed within the spectral span of 4000–500 cm^−1^, coupled with 64 accumulations. The FTIR spectra were collected in a Vertex70 (Bruker) spectrometer with a Golden Gate Single Reflection Diamond ATR System accessory. Thermogravimetric analysis (TGA): TGA analysis was performed in the Shimadzu instrument model TGA-50, under argon flow (50 mL min^−1^), with a heating rate of 10 °C min^−1^, from 25 °C to 800 °C.

X-ray powder diffraction patterns (XRD): Powder patterns were captured employing a θ-θ transmission array, positioning the sample between two Kapton sheets, while employing a focusing mirror and a PIXcel 3D detector (operating in 1D mode) with an increment of 0.013° (2θ). These powder patterns were acquired spanning 1.5 to 5.5 degrees in 2θ, spanning a total measurement duration of 60 min. The diffractometer used was PANanalytical EMPYREAN. Transmission electron microscopy (TEM): to analyze the ordered structure of MCM-41, a transmission electron microscope from FEI Talos F200X was used. Scanning electron microscopy (SEM): Materials’ morphology was obtained with scanning electron microscopy (SEM) using a FEI microscope model Quanta 250. The samples were added to a double-sided carbon conductive tape and coated with a thin film of gold.

## 4. Conclusions

The one-pot method showed promising results, allowing a faster synthesis, and provided an alternative approach to those reported in several studies where multi-step procedures were used to obtain ordered mesoporous silicas. At the same time, this synthesis allowed a fast heating process and a strong penetrability, promoting the effective combination of silicon species and surfactants at the interface, showing that it is possible to optimize some conditions when synthesizing MCM-41. It was verified that with the synthesis and removal of the surfactant with microwave irradiation, in a time of 36 min, it was possible to obtain MCM-41 silica. The particles obtained and the textural and structural properties confirm the efficiency of the synthesis. In addition, this synthesis has not yet been reported in the literature, bringing an alternative approach to those previously reproduced in several studies where multi-step and complex procedures were used to obtain silicas with good physicochemical properties.

## Figures and Tables

**Figure 1 molecules-29-00460-f001:**
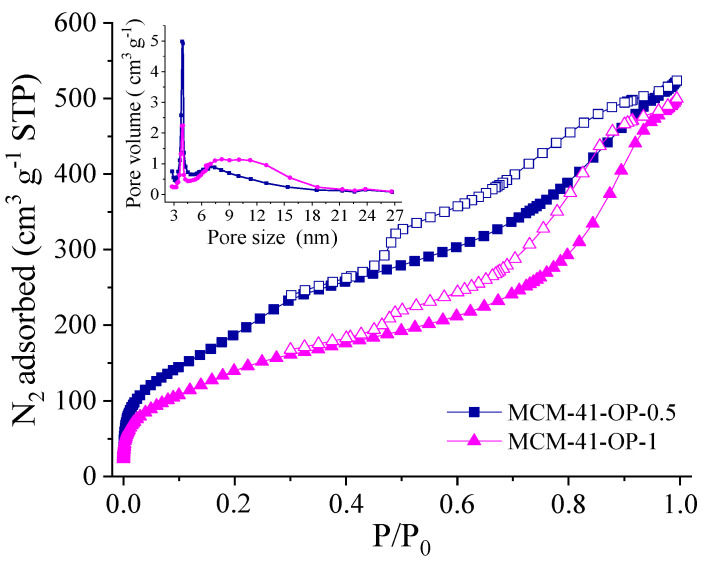
N_2_ adsorption/desorption isotherms and pore size distribution profile of the MCM-41-OP-0.5 and MCM-41-OP-1.

**Figure 2 molecules-29-00460-f002:**
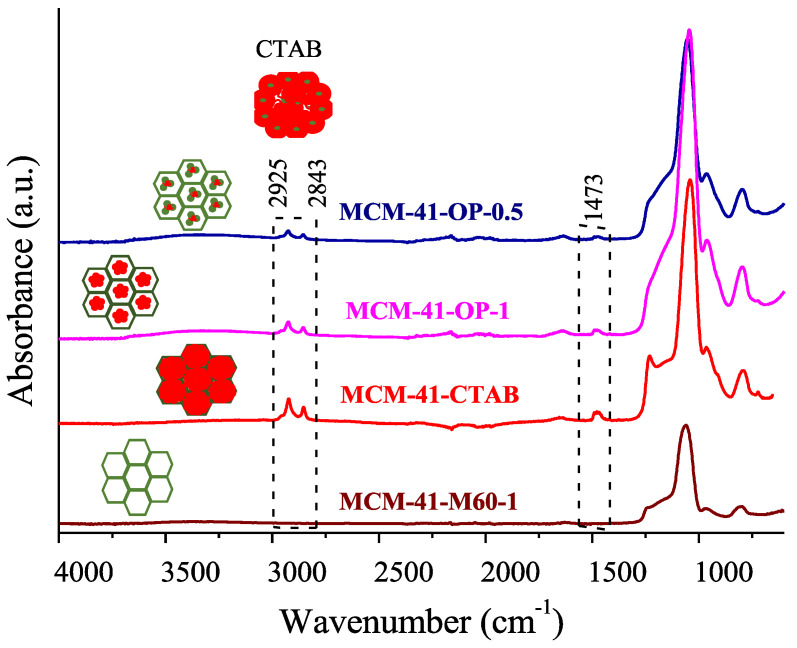
FTIR spectrum of silicas MCM-41-OP-0.5, MCM-41-OP-1, MCM-41-60-1, and MCM-41-CTAB.

**Figure 3 molecules-29-00460-f003:**
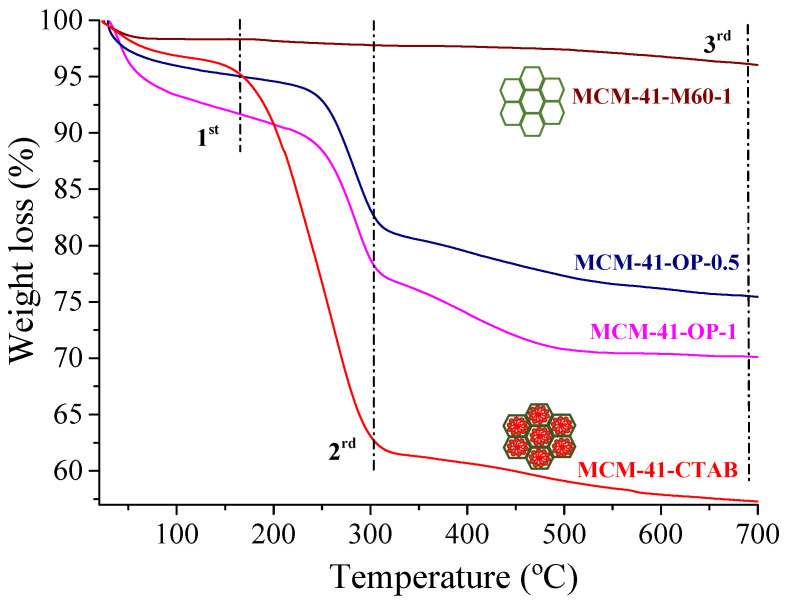
Thermograms of MCM-41 silicas before and after extraction of CTAB.

**Figure 4 molecules-29-00460-f004:**
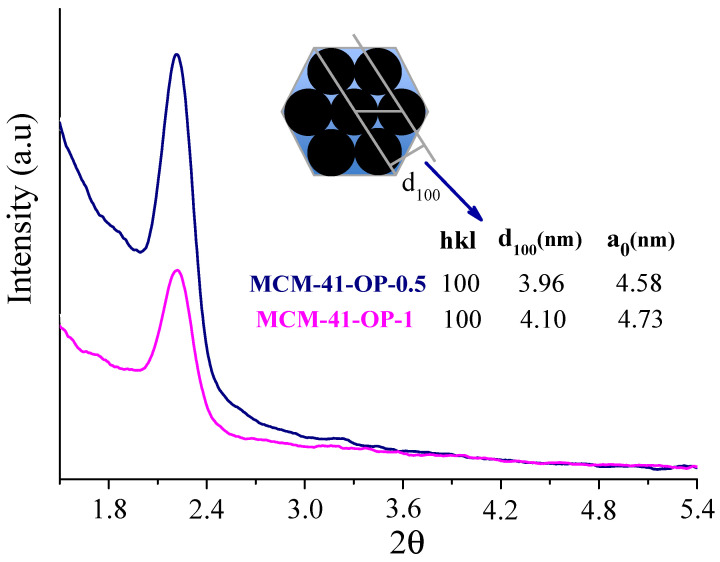
X-ray diffraction profile of MCM-41-OP-0.5 and MCM-41-OP-1.

**Figure 5 molecules-29-00460-f005:**
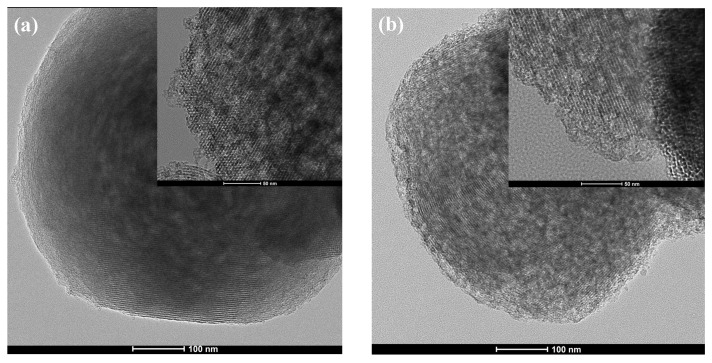
TEM images of the one-pot silicas (**a**) MCM-41-OP-0.5 and (**b**) MCM-41-OP-1.

**Figure 6 molecules-29-00460-f006:**
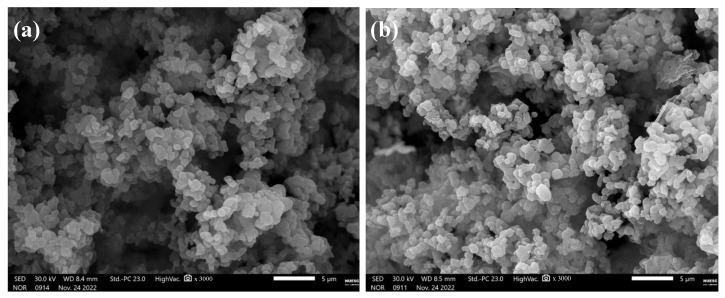
SEM images of the one-pot mesoporous silica (**a**) MCM-41-OP-0.5 and (**b**) MCM-41-OP-1.

**Figure 7 molecules-29-00460-f007:**
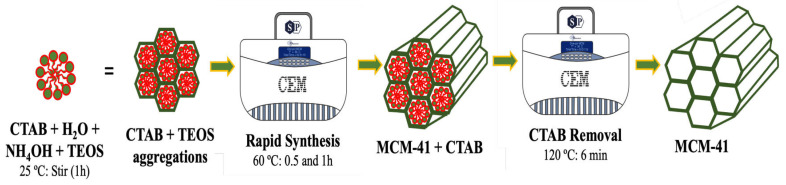
Proposed synthesis mechanism for MCM-41 one-pot mesoporous silica.

**Table 1 molecules-29-00460-t001:** Texture parameters and CHN elemental analysis of MCM-41 one-pot silicas synthesized in a microwave reactor.

Samples	N_2_ Adsorption/Desorption					CHN		
	S_BET_	Vp	V_mp t-plot_	Dp	a_0_	C	H	N
	(m^2^ g^−1^)	(cm^3^ g^−1^)	(cm^3^ g^−1^)	(nm)	(nm)	%	%	%
MCM-41-OP-0.5	760	0.77	0.15	4.04	4.54	14.1	3.2	0.7
MCM-41-OP-1	537	0.72	0.05	5.40	4.73	15.2	3.4	0.8

**Table 2 molecules-29-00460-t002:** Bibliographic survey of methodologies for MCM-41 silica.

A_BET_ (m^2^ g^−1^)	V_p_ (cm^3^ g^−1^)	Hydrothermal Treatment	CTAB Removal	References
1182	1.11	Domestic MW: 10 min—80 °C	Muffle furnace: 550 °C—4 h	[2]
1138	-	MW MARs 6: 90 min—100 °C	Muffle furnace: 550 °C—6 h	[7]
740	0.26	Conventional: 120 min—25 °C	Stirred in ethanol: 25 °C—15 min	[10]
1210	-	MW CEM: 15 min—100 °C	Muffle furnace: 500 °C—5 h	[11]
999	0.89	MW CEM: 60 min—60 °C	Muffle furnace: 550 °C—6 h	[13]
744	0.84	MW CEM: 60 min—80 °C	Muffle furnace: 550 °C—6 h	
575	0.76	Conventional: 12,100 min—30 °C	Muffle furnace: 520 °C—6 h	[14]
774	0.55	Domestic MW: 150 min—100 °C	Muffle furnace: 550 °C—10 h	[25]
**536**	**0.72**	**MW CEM:** **60 min—60 °C**	**MW CEM:** **6 min—120 °C**	**This work**
**760**	**0.77**	**MW CEM:** **30 min—60 °C**	**MW CEM:** **6 min—120 °C**	**This work**

## Data Availability

All data and materials generated or analyzed during this study are included in this article.

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
