# Peer review of "One-Pot Synthesis and Surfactant Removal from MCM-41 Using Microwave Irradiation"

_molecules, 2024, doi:10.3390/molecules29020460_

Round 1

Reviewer 1 Report

Comments and Suggestions for Authors

One-pot synthesis and surfactant removal from MCM-41 by microwave irradiation

In this paper titled “One-pot synthesis and surfactant removal from MCM-41 by microwave irradiation” the authors have used microwave radiation as an effective technique for one-pot synthesis and surfactant removal from MCM-41 silica. .  MCM-41 silica has surfactants which must be removed for their utilization.  Structural, chemical, morphological, and surface characteristics were done focusing on the impact of synthesis time. This method is efficient and also consumes less time as point out by authors. Process optimization was given importance in this work. Figures are satisfactory. References are quite relevant and recent. English is ok.

The authors are invited to address the following queries.

1.      The results and discussion part is found before the Materials and Methods. The authors shall clarify it and make sure whether it is compatible with the journal guidelines.

2.      The authors have stated that The efficiency of surfactant extraction depends on the solvent used and the strength of the interaction between the organic molecules and the structure. Is it possible to use some other solvent?

3.      What factors control the strength of the interaction between the organic molecules and the structure? Do any other processes like sonication or agitation time or temperature change this?

4.      Does the smaller amount of surfactant cause any other change in the synthesized MCM-41 silica?

5.      Does the less time consumption during synthesis produce any other change in the property of the MCM-41 silica?

The authors shall add limitation of the study, if any.

Author Response

Reviewer 1

In this paper titled “One-pot synthesis and surfactant removal from MCM-41 by microwave irradiation” the authors have used microwave radiation as an effective technique for one-pot synthesis and surfactant removal from MCM-41 silica. MCM-41 silica has surfactants which must be removed for their utilization.  Structural, chemical, morphological, and surface characteristics were done focusing on the impact of synthesis time. This method is efficient and also consumes less time as point out by authors. Process optimization was given importance in this work. Figures are satisfactory. References are quite relevant and recent. English is ok.

The authors are invited to address the following queries.

  1. The results and discussion part is found before the Materials and Methods. The authors shall clarify it and make sure whether it is compatible with the journal guidelines.

The results and discussion topic are listed according to the journal template, following the rules of the instructions for authors.

  1. The authors have stated that The efficiency of surfactant extraction depends on the solvent used and the strength of the interaction between the organic molecules and the structure. Is it possible to use some other solvent?

Yes, it is possible to use other solvents. Some studies also report the use of methanol, using sonication or Soxhlet (https://doi.org/10.1016/j.seppur.2010.11.029, https://doi.org/10.1039/C3GC40474A ). Furthermore, in this work, some tests were carried out with methanol, however the dielectric constant of methanol is high, causing overheating to occur in the microwave reactor, making it impossible to complete the tests.

  1. 3. What factors control the strength of the interaction between the organic molecules and the structure? Do any other processes like sonication or agitation time or temperature change this?

Yes, processes such as sonication or soxhlet extraction have a great influence on surfactant extraction. Furthermore, solvent, agitation time and temperature are essential factors in the process, as finding the best conditions and aligning with the best removal technique will result in more efficient removal. A better discussion was added in the manuscript version. The changes in the revised manuscript are highlighted in yellow.

  1. 4. Does the smaller amount of surfactant cause any other change in the synthesized MCM-41 silica?

The smaller amount of surfactant would allow a greater amount of micropores/pores available on the silica surface.

  1. Does the less time consumption during synthesis produce any other change in the property of the MCM-41 silica?

The shorter synthesis time can generate changes in the surface area and pore volume, as well as in the structure of the silica. However, these changes can also cause improvements in the properties of the materials, as even with a drastic reduction in time, the properties remain similar. These discussions are reported in lines 162, 180, 187-189, 198, 204, 211 and in the paragraph on line 219.

The authors shall add limitation of the study, if any. (The work has no limitations.)

Reviewer 2 Report

Comments and Suggestions for Authors

The manuscript entitled “One-pot synthesis and surfactant removal from MCM-41 by microwave irradiation”, Molecules-2802359, concerns the preparation of MCM-41 porous silica using microwave irradiation and the removal of CTAB surfactant in the same process (one-step or one-pot process). This is to minimize the time and energy used for synthesis.
Remarks
1) Lines 119-120 (comment to Table 1):
"..., since this decrease in surface area is related to the amount of surfactant still present on the silica surface, causing a blockage in the pores"
Due to the sentence above, there are some doubts about Table 1:
(i) Please explain why the sample marked MCM-41-OP-0.5 (14.1% C, lower carbon content) had a specific surface area higher than the sample marked MCM-41-OP-1 (15.2% C, higher carbon content) - 760 m2/ g vs. 537 m2/h.
(ii) Please explain why the sample marked MCM-41-OP-0.5 with a shorter time (0.5 h) of microwave radiation treatment contained 14.1% of carbon, while the MCM-41-OP-1 sample with a longer microwave radiation treatment time (1 hour) contained 15.2% carbon.
2) Line 279 in Materials and Methods:
The authors write: "These powder patterns were acquired spanning 0.5 to 10 degrees in 2..." and Figure 4 presents results only in the range of 1.4-2.5.
Please complete the remaining data in Figure 4 or explain why these data have not been included.
3) Please explain whether the proposed method of removing the surfactant from the porous silica produced allows for its recycling (controlled removal).

Author Response

Reviewer 2

The manuscript entitled “One-pot synthesis and surfactant removal from MCM-41 by microwave irradiation”, Molecules-2802359, concerns the preparation of MCM-41 porous silica using microwave irradiation and the removal of CTAB surfactant in the same process (one-step or one-pot process). This is to minimize the time and energy used for synthesis.

  1. Lines 119-120 (comment to Table 1): "..., since this decrease in surface area is related to the amount of surfactant still present on the silica surface, causing a blockage in the pores" Due to the sentence above, there are some doubts about Table 1: (i) Please explain why the sample marked MCM-41-OP-0.5 (14.1% C, lower carbon content) had a specific surface area higher than the sample marked MCM-41-OP-1 (15.2% C, higher carbon content) - 760 m2/ g vs. 537 m2/h. (ii) Please explain why the sample marked MCM-41-OP-0.5 with a shorter time (0.5 h) of microwave radiation treatment contained 14.1% of carbon, while the MCM-41-OP-1 sample with a longer microwave radiation treatment time (1 hour) contained 15.2% carbon.

The greater the amount of carbon on the surface of the silica, the greater the blockage of the pores, consequently the smaller the surface area, since these pores are not available. Regarding the synthesis time, during microwave irradiation, the heating is faster and more homogeneous, and with the rapid rise in temperature in a shorter time, the formation of impurities or unwanted phases are minimized, contributing to obtaining a material with better properties.

  1. Line 279 in Materials and Methods: The authors write: "These powder patterns were acquired spanning 0.5 to 10 degrees in 2..." and Figure 4 presents results only in the range of 1.4-2.5. Please complete the remaining data in Figure 4 or explain why these data have not been included.

We apologize for the inconvenience. The methodology was wrong. The changes in the revised manuscript are highlighted in yellow. Furthermore, a new diffractogram was added to the manuscript version.

  1. Please explain whether the proposed method of removing the surfactant from the porous silica produced allows for its recycling (controlled removal).

To determine whether the proposed method allows the recycling of porous silica, it is important to evaluate the efficiency of surfactant removal, as well as the impact on the structural and textural properties of the material. It was verified that the removal method is efficient in preserving the porous structure, however, some tests still need to be carried out to prove whether the porous silica can be recycled for subsequent applications.

Reviewer 3 Report

Comments and Suggestions for Authors

This work focuses on the extraction of surfactant from MCM-41 by microwave treatment. The researchers used several characterization methods to determine the structural, textural, and also morphological properties of their samples. The obtained results are very interesting where the researchers have clearly shown the effectiveness and speed of their treatment on the surfactant extraction. This method of preparation also makes it possible to generate the formation of a material having a high density of silanol compared to those calcined which is a key factor for functionalizing them with organosilane (for example APTES). In general, the work is well carried out, and the discussions are logical, but there are a few points that need to be improved before this paper is accepted. For this, I recommend minor revisions.

Comments

1) It is very important to add the nitrogen sorption isotherm for the calcined MCM-41 and to compare them with those that were modified by microwave.

2) please cite those references that are relevant to this topic;

https://doi.org/10.1007/s11696-017-0279-4, https://www.mdpi.com/2073-4344/11/2/219, https://doi.org/10.1016/j. apcata.2014.10.022, https://doi.org/10.1007/s10904-020-01689-1, https://doi.org/10.1016/j.arabjc.2013.07.049

3) The XRD spectrum should be carried out between 2theta = 1-6° to observe the other peaks as well as to confirm the good organization of the structure.

4) It is fascinating to calculate and compare the mesh parameter a0.

Author Response

Reviewer 3

This work focuses on the extraction of surfactant from MCM-41 by microwave treatment. The researchers used several characterization methods to determine the structural, textural, and also morphological properties of their samples. The obtained results are very interesting where the researchers have clearly shown the effectiveness and speed of their treatment on the surfactant extraction. This method of preparation also makes it possible to generate the formation of a material having a high density of silanol compared to those calcined which is a key factor for functionalizing them with organosilane (for example APTES). In general, the work is well carried out, and the discussions are logical, but there are a few points that need to be improved before this paper is accepted. For this, I recommend minor revisions.

  1. It is very important to add the nitrogen sorption isotherm for the calcined MCM-41 and to compare them with those that were modified by microwave.

The adsorption isotherm and transmission micrograph have been added in the supplementary material. In the manuscript, a discussion comparing the materials was included. The changes in the revised manuscript are highlighted in yellow.

2) please cite those references that are relevant to this topic;

https://doi.org/10.1007/s11696-017-0279-4, https://www.mdpi.com/2073-4344/11/2/219, https://doi.org/10.1016/j. apcata.2014.10.022, https://doi.org/10.1007/s10904-020-01689-1, https://doi.org/10.1016/j.arabjc.2013.07.049

Thanks for the suggestions. Some references were added to the manuscript version.

  1. The XRD spectrum should be carried out between 2theta = 1-6° to observe the other peaks as well as to confirm the good organization of the structure.

A new XRD spectrum has been added to the manuscript version. Furthermore, due to the methodology used in the work, only the main peak referring to the hexagonal mesoporous structure was noted in these samples.

  1. It is fascinating to calculate and compare the mesh parameter a0.

The values of the a0 parameter were calculated and presented in Table 1 and Figure 4. Furthermore, a discussion was added in lines 199-203.

Reviewer 4 Report

Comments and Suggestions for Authors

The article titled "One-pot synthesis and surfactant removal from MCM-41 by microwave irradiation” directed to apply the microwave irradiation as an innovative strategy for one-pot synthesis and surfactant elimination (cetyltrimethylammonium bromide - CTAB) from MCM-41, introducing a rapid and efficient methodology. The researcher claimed that this method remove surfactant challenge along with accelerating the synthesis process. The work is novel and industry relevant. The comments are as follows:

1.       The introduction is well drafted. However, addition of limitations of other techniques will improve the manuscript.

2.       Line 109: 1300 m2 g-1, is this correct unit representation?

3.       Line 117: Results obtaneid from Oliveira et al. 2022[13]. Check the spelling of obtained. What does mean by obtained? Taken from ref 13. (if permission is required needs to be obtained)

4.       Materials section: add country of material supplier or producer

Author Response

Reviewer 4

The article titled "One-pot synthesis and surfactant removal from MCM-41 by microwave irradiation” directed to apply the microwave irradiation as an innovative strategy for one-pot synthesis and surfactant elimination (cetyltrimethylammonium bromide - CTAB) from MCM-41, introducing a rapid and efficient methodology. The researcher claimed that this method remove surfactant challenge along with accelerating the synthesis process. The work is novel and industry relevant. The comments are as follows:

  1. The introduction is well drafted. However, addition of limitations of other techniques will improve the manuscript.

The introduction has been modified considering the suggestion of the reviewer. And the introduced changes in the revised manuscript were highlighted in yellow.

  1. Line 109: 1300 m2 g-1, is this correct unit representation?

The units were placed in envelope.

  1. Line 117: Results obtained from Oliveira et al. 2022[13]. Check the spelling of obtained. What does mean by obtained? Taken from ref 13. (if permission is required needs to be obtained)

The word obtained was removed from the manuscript. The data are references from another article, so the way it was presented in the manuscript was modified.

  1. Materials section: add country of material supplier or producer.

The country was added to the manuscript.
